# Parrotfish corallivory on stress-tolerant corals in the Anthropocene

**Víctor Huertas**[1,2]*, **Renato A. Morais**[1,2], **Roberta M. Bonaldo**[3], **David R. Bellwood**[1,2]

**1** Research Hub for Coral Reef Ecosystem Functions, College of Science and Engineering, James Cook University, Townsville, Queensland, Australia, **2** ARC Centre of Excellence for Coral Reef Studies, James Cook University, Townsville, Queensland, Australia, **3** Grupo de História Natural de Vertebrados, Instituto de Biologia, Universidade Estadual de Campinas, Campinas, São Paulo, Brazil

* victor.huertas@my.jcu.edu.au

**Data Availability Statement:** All data files are available from the Research Data JCU data management platform (DOI: 10.25903/g9am-9w70).

## Abstract

Cumulative anthropogenic stressors on tropical reefs are modifying the physical and community structure of coral assemblages, altering the rich biological communities that depend on this critical habitat. As a consequence, new reef configurations are often characterized by low coral cover and a shift in coral species towards massive and encrusting corals. Given that coral numbers are dwindling in these new reef systems, it is important to evaluate the potential influence of coral predation on these remaining corals. We examined the effect of a key group of coral predators (parrotfishes) on one of the emerging dominant coral taxa on Anthropocene reefs, massive *Porites*. Specifically, we evaluate whether the intensity of parrotfish predation on this key reef-building coral has changed in response to severe coral reef degradation. We found evidence that coral predation rates may have decreased, despite only minor changes in parrotfish abundance. However, higher scar densities on small *Porites* colonies, compared to large colonies, suggests that the observed decrease in scarring rates may be a reflection of colony-size specific rates of feeding scars. Reduced parrotfish corallivory may reflect the loss of small *Porites* colonies, or changing foraging opportunities for parrotfishes. The reduction in scar density on massive *Porites* suggests that the remaining stress-tolerant corals may have passed the vulnerable small colony stage. These results highlight the potential for shifts in ecological functions on ecosystems facing high levels of environmental stress.

## Introduction

The scale and severity of disturbances that coral reefs have endured in the last decade have altered their coral assemblages and caused profound changes to their composition and physical appearance [1, 2]. By 2020, the footprint of severe tropical storms and coral bleaching—two of the primary manifestations of climate change on coral reefs—have left an indelible mark on reefs throughout the tropics [3, 4]. Temperature-induced coral bleaching, the leading cause of coral mortality, has been associated with recent episodes of widespread loss of coral cover on the Great Barrier Reef [5], and around the globe [6–9]. Furthermore, the uneven susceptibility

**Funding:** This work was supported by a James Cook University Postgraduate Research Scholarship (V.H.), the Australian Research Council (grants CE140100020 and FL190100062 to D.B.), and a Lizard Island Doctoral Fellowship from the Lizard Island Reef Research Foundation (R.M.). The funders had no role in study design, data collection and analysis, decision to publish, or preparation of the manuscript.

**Competing interests:** The authors have declared that no competing interests exist.

of different coral taxa to stressors has driven extensive coral community changes. On the Great Barrier Reef, for example, an increase in the proportion of massive *Porites* has been reported relative to the remaining coral cover [1, 10].

Changes in coral cover following acute disturbances are quickly apparent [4], but reef degradation also has knock-on effects across the entire ecosystem [11, 12]. Reef fish communities, for example, respond in complex ways to shifts in coral communities [12–20]. Typically, these responses have been investigated from the perspective of fish abundance and community structure while the impact on the capacity of fishes to deliver key ecological functions has received less attention [21]. This is important because the impact of climate change on coral reefs is expected to escalate [2, 22, 23]. Thus, as coral reefs cope with a warming ocean, a critical question emerges: will the delivery of ecological functions by fishes change on Anthropocene reefs? In this study, we investigate this question by focusing on parrotfish predation on corals.

Although parrotfishes generally feed on algal turf-covered substrata, they also occasionally scrape the surface of live corals [24–27]. On the Great Barrier Reef, multiple parrotfish species have been reported to bite on massive *Porites*, including *Scarus flavipectoralis*, *S. niger*, *S. frenatus*, *S. rivulatus*, *Chlorurus microrhinos*, *C. spilurus*, *Cetoscarus ocellatus*, and *Bolbometopon muricatum* [24, 28, 29]. Indeed, parrotfish corallivory may, under specific circumstances, compromise the survival of corals, regulating their distribution and abundances [24, 25, 30–32]. Bonaldo and Bellwood (2011) provided the first quantitative assessment of parrotfish predation on massive *Porites* on the Great Barrier Reef (GBR). This study highlighted that parrotfish corallivory on the GBR primarily affects massive *Porites* colonies [24]. Welsh et al. [33] provided further evidence for this negative impact showing that clustered bites can trigger partial mortality in *Porites* corals. Since these studies were published, however, coral cover at Lizard Island has decreased and strong compositional changes have been observed [1, 34]. These changes may have impacted the extent of parrotfish predation on corals, potentially endangering the remaining massive *Porites* colonies.

The purpose of this study, therefore, was to investigate the footprint of parrotfish predation on massive *Porites* on Anthropocene reefs, where the proportion of substratum occupied by turf algae (their primary feeding microhabitat) is expected to have increased while most corals, except *Porites*, have become less abundant (especially acroporid corals) [34]. Specifically, we addressed two questions:

- What is the distribution of parrotfish predation on massive *Porites* following reef degradation?

- Has reef degradation affected the intensity of parrotfish predation?

## Materials and methods

We conducted this study on the coral reef between Palfrey and South islands (S 14˚ 41' 57", E 145˚ 26' 55"), just south of Lizard Island (Fig 1A), on the northern Great Barrier Reef (GBR). In the last decade, this reef has been affected by two cyclones (Ita in 2014 and particularly Nathan in 2015) and two consecutive mass bleaching episodes (2016 and 2017) [34], making it an appropriate location for investigating the effect of reef degradation on fish-coral interactions. Importantly, fishing in the area is prohibited [35] and because it is located in the middle of the continental shelf, its exposure to land-based sediment inputs is limited [36].

We selected four reef zones that differ in depth and wave exposure following [24]: slope (7–10 m deep), crest (0.7–2 m), flat (0.5–1 m), and back reef (5–8 m) (Fig 1A). To quantify parrotfish predation on massive *Porites*, we counted the number of parrotfish scars on massive

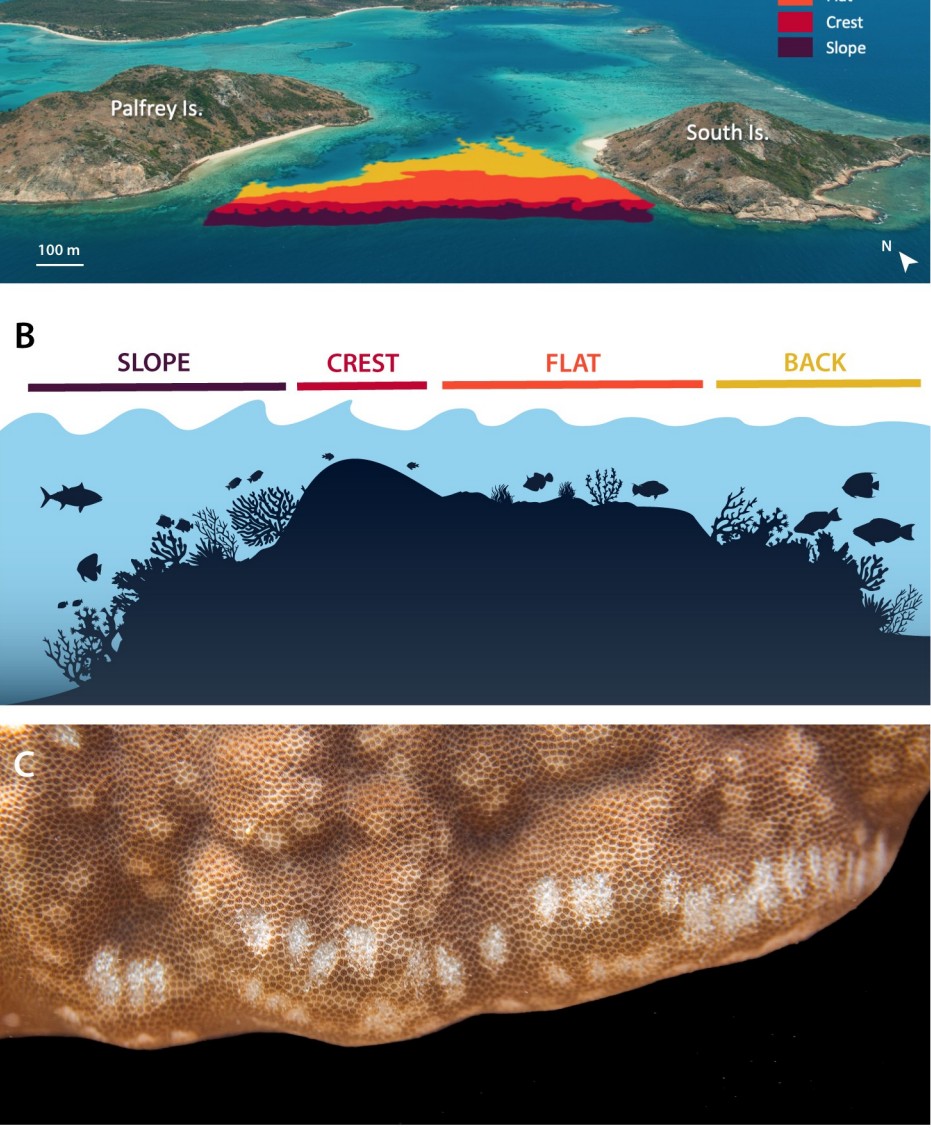

**Fig 1.** Study location between Palfrey and South islands, in the Lizard Island group, Great Barrier Reef (A), illustration representing the reef habitats across the reef profile at the study location (B), and a photograph showing a series of *Scarus* parrotfish scars on a massive *Porites* coral (C). Photographs taken by Victor Huertas.

*Porites* (Fig 1B). Parrotfish scars were distinguished by a pair of opposing oval-shaped marks on the coral surface with shape and depth varying depending on the size of the fish. The scars left by different parrotfish species cannot be separated except in separating whether they were inflicted by a scraper or an excavator, with excavators producing wider and deeper scars than scrapers [24]. In each reef zone we collected three sets of data to evaluate: a) the composition of the benthic substratum, b) parrotfish abundances, and c) the number of parrotfish scars on massive *Porites*. This study was conducted under approval from James Cook University's

Animal Ethics Committee (Ethics permit #A2627) and a Great Barrier Reef Marine Park Authority research permit (Permit #G17/38142.1).

## Benthic composition

We quantified the benthic composition using a photoquadrat method following the methodology used by [24]. A minimum of seven 20 m transects were laid in each reef zone (slope, crest, flat, and back reef; 41 transects). Images were taken from a distance of approximately 1.5 m that subsequently permitted a 1 m$^2$ quadrat to be overlaid on the image prior to analysis. In each transect, the substratum was photographed at every other meter (n = 410 frames). Next, we estimated the percent of live coral and the percent of massive *Porites* from 10 points laid on each photoquadrat (n = 4,100 points) via a stratified randomization process using the software photoQuad [37].

## Parrotfish abundance

We conducted underwater visual surveys to determine parrotfish abundances. A team of two divers conducted a series of twelve 50 m x 2 m tape transects on each of the reef zones in 2018. To avoid underestimating parrotfish abundance by scaring the fish away [38, 39], the 50 m tape was laid by the same diver simultaneously to the counts. Only those parrotfishes larger than 10 cm in length observed within 1 m on either side of the tape were recorded. The 2018 parrotfish counts were then compared with surveys using the same methods conducted on the same reef in 2008 [40], i.e., before the series of severe disturbances that affected this reef.

## Parrotfish predation on massive *Porites*

We photographed massive *Porites* colonies from above with a Nikon Coolpix W300 digital camera with a scale ruler to calibrate the measurement of their horizontal planar surface area (subsequently termed "surface area"). All images were taken from the same position (i.e., directly above the colony) to ensure consistency across all colonies sampled. Dead areas of the colony were not included in the total surface area. We then recorded all clearly visible parrotfish scars (see examples of parrotfish scars in [40]). During the image analysis, each scar was marked to avoid double-counting scars.

## Statistical data analysis

Parrotfish abundance across the reef profile was examined using a generalised linear model (GLM). To account for the overdispersion and non-normality of our count data, we fitted our data with a negative binomial regression with year (2008 vs 2018) and reef zone (slope, crest, flat, and back) as predictors. We assessed the fit and relevant assumptions of the model with residual plots, which were all satisfactory after we removed a single outlier that we considered to be an error. We also used the Akaike Information Criterion (AIC) in a model comparison framework to determine whether any subset of the predictors generated a more parsimonious model (S1 Table in S1 File). Finally, we assessed differences in parrotfish abundance pre- and post-disturbances at each zone with pairwise comparisons using Tukey-adjusted p-values (function 'emmeans' in the R package 'emmeans').

We also investigated if parrotfish predation on massive *Porites* colonies was influenced by colony size. To determine (a) if the likelihood of being bitten or not (binary) changed depending on the colony's size, and whether it varied across zones, we fitted a GLM with a binomial error distribution. Next, (b) we focused on colonies that have been bitten (i.e., with at least one scar) to investigate the influence of colony size on the magnitude of parrotfish predation. In

this case, we fitted a GLM using a negative binomial distribution to determine if there was a relationship between the number of scars on individual colonies, and colony surface area, as well as the reef zone in which they were located. Again, we performed model selection on subset of predictors based on AIC scores (S2 Table in S1 File).

Finally, (c) we also modelled the relationship between the density of scars on individual colonies and their surface area and reef zone. Here, we fitted the data using a GLM with a gamma distribution. Although density can be decomposed in its two original components (number of scars and area) we did not include surface area as an offset in the model. Offsets allow for the removal of variability that arises from confounding dimensional factors (e.g., colony size) from the response variable. They do this by dividing each observation of the response variable by the corresponding observation of the offset variable. However, in this case, colony size was our predictor of interest, rather than a scaling variable. As colony size varies both within and across the reef zones (S1 Fig in S1 File), including it as an offset would have kept it at a constant value, thus constraining our ability to detect patterns that may emerge with it.

In both cases (b and c), colony surface area was log transformed prior to analysis. A pairwise post-hoc comparison was conducted to examine variation in the number of scars among reef zones. All statistical analyses were conducted in the software R v.3.6.1 [41] using the packages *ggplot2* [42], *glmmTMB* [43], *MASS* [44], *MuMIn* [45], and *emmeans* [46].

## Results

### Coral cover

Between 2008 and 2018 the study site lost two thirds of its live coral cover, which fell from 22% to ~7% (Table 1). Live coral cover declined the least on the back reef (-25.0%) vs. -72.2% on the slope, -77.4% on the flat, and -86.2% on the crest (Fig 2, Table 1). Although the area occupied by live corals declined in all four reef zones, coral loss varied by taxa and resulted in a marked increase in the proportion of massive *Porites* among hard corals on the slope (from 30.6% to 58.9% of the total coral cover), the crest (4.9% to 20.0%), and the reef flat (9.4% to 70.6%) (Fig 2). In relative terms, the proportion of all corals represented by massive *Porites* tripled. Thus, *Porites* shifted from being a minor component of the hard coral assemblage (with ~30% of all corals on the slope, <5% on the crest, and <10% on the flat) to the dominant coral in two of these three habitats (Fig 2).

### Parrotfish abundance

The number of parrotfishes counted in 2018 relative to 2008 remained relatively stable with minor changes depending on the reef zone (Fig 3A). We did not detect statistically significant differences in parrotfish abundance between the two time periods (pre and post-disturbances) on the slope or crest (GLM; slope effect size$_{[2008 \text{ vs } 2018]}$ = 1.63, CI$_{95}$ = 0.78–2.48, p = 0.063; crest effect size$_{[2008 \text{ vs } 2018]}$ = 1.47, CI$_{95}$ = 0.73–2.20, p = 0.130) (S2 Table in S1 File), however, the number of parrotfish was lower on the reef flat in 2018 compared to 2008 (GLM; effect size$_{[2008 \text{ vs } 2018]}$ = 3.50, CI$_{95}$ = 1.75–5.25, p < 0.001, S2 Table in S1 File) but was higher on the back reef in 2018 compared to 2008 (GLM; effect size$_{[2008 \text{ vs } 2018]}$ = 0.50, CI$_{95}$ = 0.25–0.75, p = 0.007, S2 Table in S1 File).

### Parrotfish predation on massive *Porites*

Following the impact of cyclones Ita and Nathan, and the 2016 and 2017 mass coral bleaching events, we observed a marked reduction in the density of parrotfish scars on massive *Porites* at the study site in 2018 compared to 2008. This was mainly driven by a precipitous decline in

**Table 1. Percentage of live hard coral cover in 2008 and 2018 at the study site, Lizard Island, on the Great Barrier Reef.** Data for 2008 sourced from [24].

| | Massive *Porites* | | | Total live coral | | |
|---|---|---|---|---|---|---|
| | **2008** | **2018** | **Change** | **2008** | **2018** | **Change** |
| **Slope** | 7.31 | 3.91 | -46.54 | 23.89 | 6.64 | -72.22 |
| **Crest** | 1.77 | 1.00 | -43.64 | 36.14 | 5.00 | -86.16 |
| **Flat** | 0.65 | 1.09 | 69.09 | 6.83 | 1.55 | -77.38 |
| **Back** | 15.59 | 6.75 | -56.71 | 21.12 | 15.83 | -25.02 |
| *Average* | *6.33* | *3.19* | *-49.65* | *21.99* | *7.25* | *-67.02* |

coral scar densities on the crest and the flat in 2018, the habitats that exhibited the vast majority of parrotfish scars in 2008 [24]. While these shallow habitats continued to support the largest densities of scars in 2018, parrotfish coral scars were only 40% and 25% of the 2008 values on the crest and the flat, respectively (Fig 3B, S3 Table in S1 File). The number of scars on the slope and the back reef in 2018 was slightly higher than in 2008, but these zones only accounted for 20.5% and 12.6% of the total number of scars present on the reef in 2018, respectively. To investigate the reason for this apparent reduction in parrotfish predation on massive *Porites* in 2018 (with limited overall change in parrotfish density), we looked at the potential effect of colony size on the pattern observed.

Across the four reef zones, the probability of a *Porites* colony being bitten by a parrotfish in 2018 did not vary depending on colony size (i.e., individual coral surface area available for predation) (S4 Table in S1 File). Parrotfishes, therefore, did not choose whether to bite corals or not based on coral colony size. Among colonies that had scars, however, there was a clear colony size effect (Fig 4).

We found a strong positive relationship between the number of scars and the colony surface area on the slope (GLM Estimated coefficient = 0.65 ± 0.17), crest (GLM Estimated coefficient = 1.04 ± 0.31), and back reef (GLM Estimated coefficient = 1.44 ± 0.22) (Fig 4A). We did not detect a relationship on the reef flat (GLM Estimated coefficient = 0.04 ± 0.37). Overall, of those colonies that had scars the number of scars was generally higher on larger colonies. However, the most revealing values were scar densities. We found a strong negative relationship between the density of scars and the *Porites* colony surface area for all four reef zones (Fig 4B, S5 Table in S1 File). Thus, the density of scars was far higher on small colonies and diminished rapidly as the size of the colony increased. This negative relationship was consistent across all four reef habitats, although it varied slightly in magnitude.

## Discussion

Between 2008 and 2018, our Lizard Island study site lost two thirds of its coral cover, with live corals covering only ~7% of the substratum in 2018. During this period, the forereef habitat (i.e., slope and crest) was affected the most, with a coral cover decline of up to 86%. These results are congruent with other studies conducted at the same (exact) location [10, 34, 47] and reflect the broader declines reported from the Great Barrier Reef [5]. Most of the coral mortality in the forereef was caused by damage from cyclone-generated waves, especially from cyclone Nathan in 2015 [47], but also by the severe bleaching events that followed in 2016 and 2017. The corals most heavily affected were habitat-forming *Acropora*, while keystone reef-builders like massive *Porites* endured the severe weather well [34]. This uneven sensitivity of coral taxa to stressors resulted in an increase in the proportion of massive *Porites* in 2018, an effect that has been previously reported in other locations [48, 49] and this stems from the

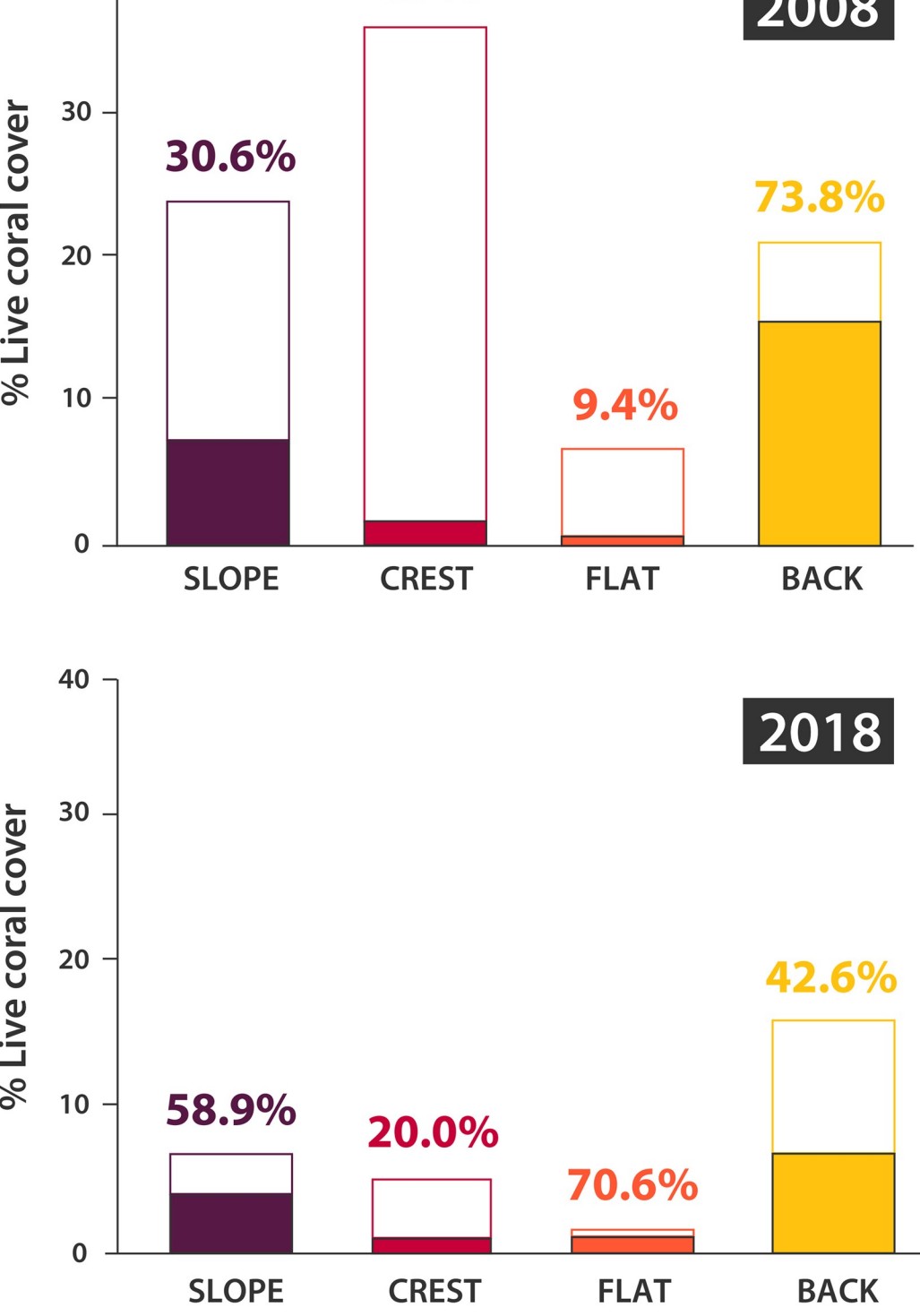

**Fig 2. Changes in the benthic cover at Lizard Island, on the Great Barrier Reef.** Percentage of reef substratum occupied by massive *Porites* (filled) and other hard corals (empty) in 2008 and 2018. Purple, red, orange, and yellow indicate the slope, crest, flat, and back reef habitats, respectively. Values above the bars indicate the percentage of massive *Porites* cover relative to total coral cover for the respective habitat.

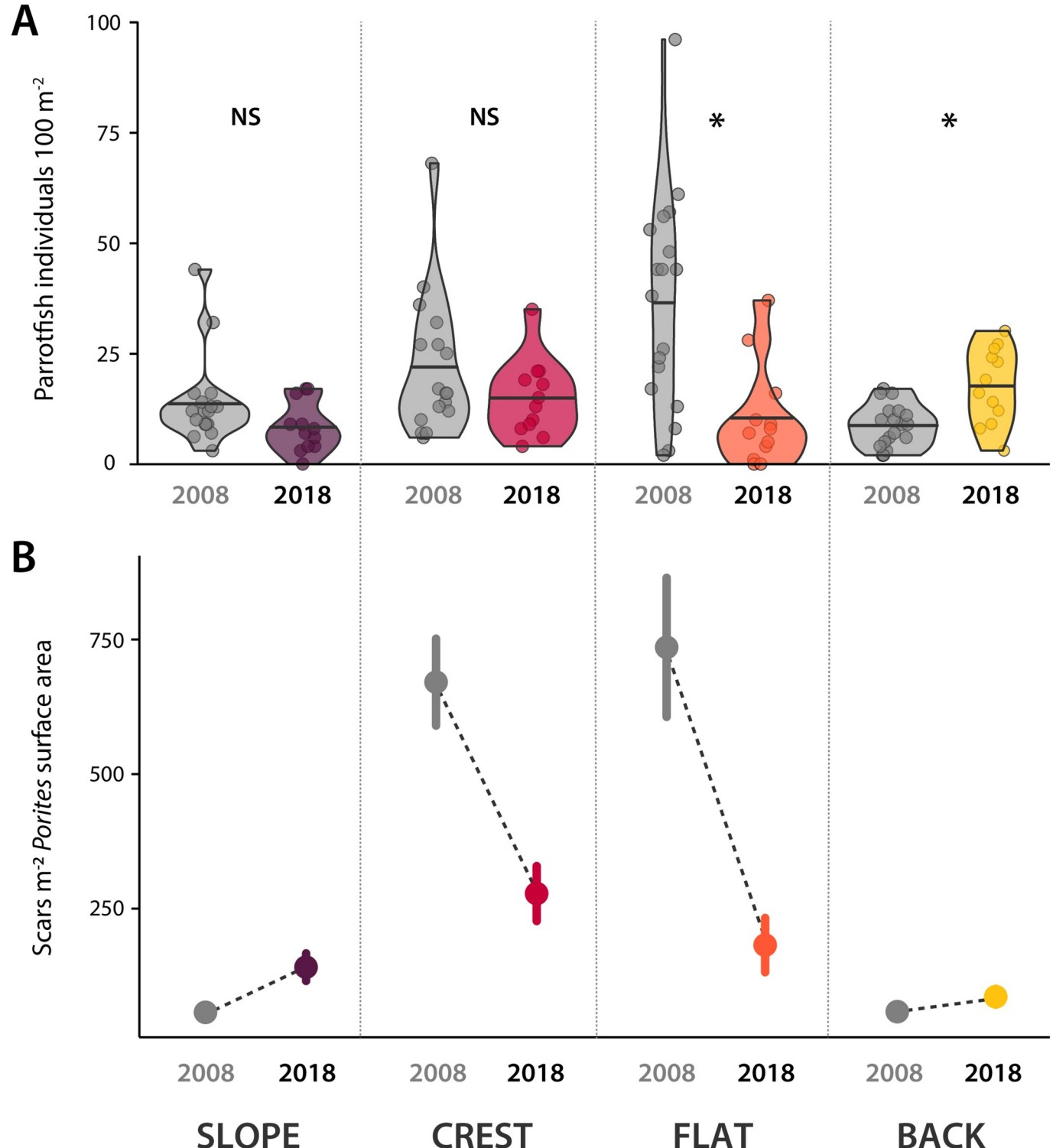

**Fig 3. Parrotfish abundance and predation pressure on corals across the study location at Lizard Island, on the Great Barrier Reef.** (A) Violin plots representing parrotfish abundance in 2008 (grey) and 2018 (coloured). Horizontal lines indicate the mean. Circles are individual samples. NS: statistically non-significant; *: statistically significant. (B) Predation pressure on massive *Porites* on the slope, crest, flat and back reef zones at the study site in 2008 (grey) and 2018 (coloured). Circles represent the mean number of scars m⁻² and lines indicate the standard error of the mean. Data from 2008 were sourced from [24]. Purple, red, orange, and yellow indicate the slope, crest, flat, and back reef habitats, respectively.

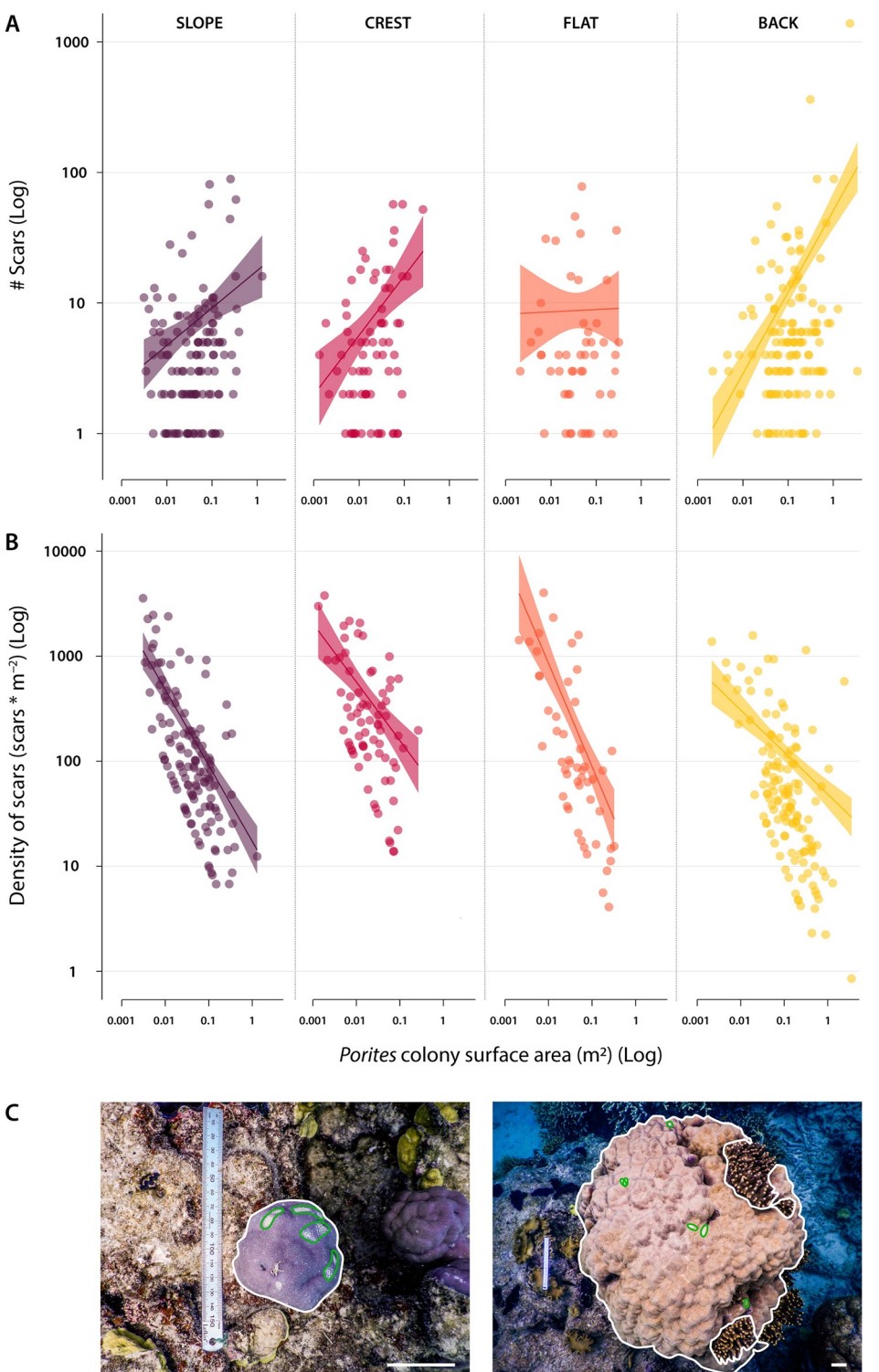

**Fig 4. Parrotfish predation pressure on massive *Porites* at Lizard Island, Great Barrier Reef.** (A) The relationship between the number of scars and the size of massive *Porites* colonies. Fitted lines and bands are, respectively, generalised linear model fits and their 95% confidence interval. (B) The relationship between the density of parrotfish scars and the colony surface area. Note that axes are on a log scale and reveal the exceptionally high scar density on small colonies. (C) Examples of small (left) and large (right) massive *Porites* colonies at the study site. The same ruler is featured as a scale in both images. Scars are outlined in green and planar colony surface area in white. Scalebars = 5 cm. Purple, red, orange, and yellow indicate the slope, crest, flat, and back reef habitats, respectively.

relative high resilience of massive *Porites* to hydrodynamic forcing [10, 50] and heat stress [51–54].

Despite the loss of most other corals, a decrease in absolute massive *Porites* cover, previous evidence that parrotfishes selectively target massive *Porites* [24, 25], and stable parrotfish abundances, we did not detect an increase in parrotfish predation on massive *Porites* in 2018. Indeed, we found that the number of bites appears to have declined relative to 2008. The apparent reduction in predation pressure is best illustrated by the difference in the overall density of scars. Considering, for example, a typical section of reef from the study site (composed of 35.2% reef flat, 27.3% of back reef, 24.2% of reef slope, and 13.3% of crest; as measured from satellite images, ESM Methods), using the *Porites* cover (Table 1 and Fig 1, respectively) and assuming a uniform distribution of *Porites* colonies yields a ~40% reduction in the mean density of scars at the study location from 2008 to 2018 (from 676 to 400 scars per 100 m$^2$ of massive *Porites* planar surface area). We offer two hypotheses that may explain this phenomenon.

## Changes in parrotfish abundance or behaviour in response to reef degradation

A decrease in parrotfish abundance after the disturbances could have contributed to a decline in parrotfish scars [30]. However, the major disturbances had little effect on overall parrotfish abundances. One possible explanation for this lack of a response is that foraging parrotfishes predominantly feed on non-coral surfaces covered in algal turfs [28]; coral-feeding is relatively infrequent [25, 55, 56]. Indeed, parrotfishes and other herbivorous fishes usually respond positively to extensive coral mortality, typically with a rapid growth in biomass [19, 57]. This is probably because these reefs provide larger areas covered with turf algae, their preferred feeding microhabitat [19, 58–63]. With increased or maintained parrotfish populations, one may anticipate consistent or increased corallivory, if coral-feeding activity remains a constant, if small, proportion of the diet of these fishes. Instead, we observed clear evidence of decreased predation. This decrease could be attributed to a change in foraging behaviour. Most parrotfishes target turf algae-covered surfaces on the reef [56]. Thus, the observed overall reduction in scars on massive *Porites* may reflect a shift in the foraging behaviour of parrotfishes driven by an increase in the availability of turf-covered substratum following mass coral mortality [64]. While we cannot confirm that this was the case at Lizard Island, changes in the foraging behaviour of parrotfishes following coral mortality have been documented at other locations [65].

It is also worth noting that changes in reef fish populations associated with habitat loss may take some time to become apparent [66], and therefore trends in parrotfish abundance need to be interpreted with caution. Nevertheless, parrotfish scars on massive *Porites*, contrary to parrotfish populations, decreased in frequency. Thus, our data suggests that, at least in the short term, changes in the composition and structure of the reef diminished the footprint of parrotfish predation on massive *Porites*. This underscores the importance of incorporating direct measures of ecological function, such as coral predation, in ecological studies assessing responses to reef degradation. Overall, parrotfish abundance at Lizard Island does not appear to be a major factor influencing predation on corals at this time, consistent with studies in other reef systems [65, 67].

## The effect of colony size

In 2008, the highest levels of parrotfish predation were observed on the crest and the reef flat [24]. On a mid-shelf reef such as Lizard Island, these shallow reef habitats typically concentrate the largest foraging activity of parrotfishes regardless of whether they graze on turf algae

surfaces [68] or predate on live coral [24]. In 2018, after a series of severe disturbances impacted this reef, the crest and reef flat continued to sustain the highest intensity of parrotfish corallivory, albeit with much lower scar densities. These lower densities, despite minor or unclear changes in parrotfish numbers, raise the question of why the pattern of feeding remained but overall rates decreased, especially on the flat and the crest. One possible explanation is coral colony size.

The number of parrotfish scars on massive *Porites* increased with colony size. However, this increase was not proportional and, as a result, the density of scars was substantially higher on small *Porites* colonies than on bigger, older, colonies. This pattern was observed across all reef zones and, thus, appears to be intrinsic to the trophic interaction, rather than influenced by depth or exposure to waves. The higher density of scars on small colonies could be associated with the degree of surface curvature, with more curved small colonies being more prone to parrotfish predation, especially by excavating parrotfishes [28]. Alternatively, it may also reflect other properties, such as fewer nematocysts or differences in tissue thickness. Regardless of the cause, focused predation on small colonies could have important ramifications.

The reduction in the overall intensity of parrotfish predation in 2018 coupled with the higher density of scars on small colonies suggest that a shift in the size structure of massive *Porites* may have occurred. It is possible that the series of disturbances this reef experienced over the last decade may have not only driven an increase in the proportion of massive *Porites* cover relative to total coral cover, but it may also have increased the proportion of large massive *Porites* colonies via differential survival of large colonies. This is particularly relevant if we consider the patterns of coral mortality associated with tropical cyclones. Lower survivorship of juveniles is a generalised feature of corals—and indeed all other animals. In corals, structurally-complex morphotypes (e.g., tabular, corymbose corals) become increasingly more vulnerable to hydrodynamic forces as they grow in size [69]. However, massive corals are disproportionally impacted by intense wave action (such as those generated by cyclones Ita and Nathan) when they are small [50]. Thus, it is likely that the abundance of juvenile massive *Porites* decreased over the study period due to differential mortality of juveniles following the impact of cyclones Ita, and especially Nathan. This could have contributed to the decline in parrotfish scars.

The comparatively high scar densities we observed on small colonies will impose an energetic burden on this vulnerable life stage, potentially constraining colony growth by diverting resources towards wound healing [70]. Indeed, even when parrotfish corallivory does not cause total or partial colony mortality, persistent parrotfish predation may result in reduced colony growth, lower reproductive output [71], higher exposure to disease [72], or a reduced ability to cope with future environmental stress [31]. Given the high scar densities that small *Porites* colonies are exposed to, this appears to be a difficult ontogenetic phase for massive *Porites* corals, with the potential for a size-based escape from the worst effects of predation by parrotfishes.

This is important because although *Porites* colonies can handle high levels of environmental stress, high rates of parrotfish predation on small colonies may represent yet another source of colony mortality; with potential negative effects acting in synergy with other types of disturbances (e.g., storms or heat stress). As massive *Porites* become one of the dominant corals on reefs in the Anthropocene, the need to understand the role of potential natural coral predators in regulating their growth and habitat distributions increases. Our study corroborates findings from previous studies indicating that parrotfishes are important coral predators on Indo-Pacific reefs that can be responsible for significant colony damage [24, 25, 33, 73, 74]. Potential reductions in the abundance of young corals in particular is a concern as this demographic bottleneck may underpin further coral decline [75, 76].

## Conclusion

Our findings provide new insights into the effect of parrotfish corallivory on the coral reefs of the Anthropocene. Previous research indicated that parrotfishes may be capable of shaping the survival and distribution of massive *Porites* [24, 25]. Here, we showed that the overall density of scars diminished in the aftermath of severe degradation, suggesting that an increase in the relative proportion of massive *Porites* did not stimulate parrotfish corallivory, at least in the short term. Importantly, our findings indicate that the impact of parrotfish predation on massive *Porites* is disproportionately greater on small colonies. Therefore, colony growth may provide escape from predation. Although colony-size specific parrotfish predation on massive *Porites* has not been documented on undisturbed reefs, if the observed pattern of disproportionately higher scar density on small colonies is a persistent pattern, parrotfish corallivory on Indo-Pacific reefs could represent an important factor modulating the dynamics of massive *Porites* populations in the Anthropocene. Given the exceptionally long lifespan of massive *Porites* colonies, however, it is likely that the ecosystem-wide effects of high predation on their early ontogenetic stages may take several decades to emerge.

Parrotfishes have probably scarred corals for millions of years and our results suggest that they will continue to do so in the Anthropocene. Despite little evidence of strong effects of parrotfish predation on reef growth, we show that parrotfish corallivory on small colonies may be a significant factor shaping the distribution and abundance of this dominant coral on Anthropocene reefs. It is unclear what the long-term effects of coral reef degradation will be and whether parrotfish predation will inflict enough damage to have a significant impact on dwindling coral communities in the future. Nevertheless, the observed patterns and potential for negative impacts warrant a renewed look into the function of reef fishes in regulating coral community dynamics.

## Supporting information

**S1 File.**
(DOCX)

## Acknowledgments

We thank P. Narvaez, A. Siqueira, and J.P. Krajewski for assistance with field data collection; A. Hoggett, L. Vail and the Lizard Island Research Station staff for field support; C. Hemingson for assistance with data analysis; and C. Hemingson, M. Mihalitsis, A. Siqueira, R. Streit, S. Tebbett, A. Halford and N. H. Kumagai for insightful comments.

## Author Contributions

**Conceptualization:** Víctor Huertas, David R. Bellwood.

**Data curation:** Víctor Huertas, Roberta M. Bonaldo.

**Formal analysis:** Víctor Huertas, Renato A. Morais.

**Funding acquisition:** Renato A. Morais, David R. Bellwood.

**Investigation:** Víctor Huertas, Renato A. Morais.

**Methodology:** Víctor Huertas, Renato A. Morais, David R. Bellwood.

**Project administration:** Víctor Huertas.

**Resources:** Víctor Huertas, Roberta M. Bonaldo, David R. Bellwood.

**Software:** Víctor Huertas, Renato A. Morais.

**Supervision:** David R. Bellwood.

**Validation:** Víctor Huertas, Renato A. Morais.

**Visualization:** Víctor Huertas, Renato A. Morais, David R. Bellwood.

**Writing – original draft:** Víctor Huertas.

**Writing – review & editing:** Víctor Huertas, Renato A. Morais, Roberta M. Bonaldo, David R. Bellwood.

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
