## [Decision Letter · Decision Letter 0]

3 Feb 2021

PONE-D-20-34321

Reduced parrotfish corallivory on stress-tolerant corals in the Anthropocene

PLOS ONE

Dear Dr. Huertas,

Thank you for submitting your manuscript to PLOS ONE. After careful consideration, we feel that it has merit but does not fully meet PLOS ONE’s publication criteria as it currently stands. Therefore, we invite you to submit a revised version of the manuscript that addresses the points raised during the review process.

The manuscript needs to be more focussed on either exploring "overall effect of parrotfish predation on porites" or "effects of parrotfish predation on recruitment/juvenile dynamics". The arguments explored in the manuscript are too exclusive as there are multiple potential drivers for the patterns seen but they are not explored in an even-handed way. The comments made by myself as an independent reviewer plus the points made by reviewer number 1 provide a clearer picture of how and where the manuscript can be improved. 

The manuscript needs to not oversell the perceived importance of the study and stick to statements supported by evidence. There also needs to be an explicit recognition of how this study can be improved upon to provide mor econvincing evidence for the conclusions reached. Understanding the dynamics of fish-benthos interactions in such a rapidly changing environment is important but this work will have more relevance if it is more focussed, does not try to oversell its results and highlights how much room there is for further investigations.

We look forward to receiving your revised manuscript.

Kind regards,

Andrew Halford

Academic Editor

PLOS ONE

Journal Requirements:

2. In your Methods section, please provide additional location information of the study location, including geographic coordinates for the data set if available.

5. We note that Figure 1b in your submission contains satellite images which may be copyrighted.

We require you to either (a) present written permission from the copyright holder to publish this figure specifically under the CC BY 4.0 license, or (b) remove the figure from your submission:

a. You may seek permission from the original copyright holder of Figure 1b to publish the content specifically under the CC BY 4.0 license. 

b. If you are unable to obtain permission from the original copyright holder to publish this figure under the CC BY 4.0 license or if the copyright holder’s requirements are incompatible with the CC BY 4.0 license, please either i) remove the figure or ii) supply a replacement figure that complies with the CC BY 4.0 license. Please check copyright information on all replacement figures and update the figure caption with source information. If applicable, please specify in the figure caption text when a figure is similar but not identical to the original image and is therefore for illustrative purposes only.

6. Please include captions for your Supporting Information files at the end of your manuscript, and update any in-text citations to match accordingly. Please see our Supporting Information guidelines for more information: http://journals.plos.org/plosone/s/supporting-information

Additional Editor Comments:

I have reviewed the manuscript in addition to the external reviewer and I agree with the external reviewer that there are a number of issues that will need to be addressed before the manuscript can be reconsidered for publication. Reviewer 1 makes some important points about the potential for bias in the way bites were measured between different size colonies and these issues need to be addressed to ensure readers that there are unlikely to be artefacts in how measurements were taken.

I also agree that the analyses need to be consistent and if there is disagreement with reviewer 1's comments then a clear explanation should be given.

I would like to see some more detail about what parrotfish species are scraping the porites colonies - is it all species or just a small group? If a small group then should you have concentrated on counts of those species in your analysis.

While I found the manuscript to be well written my overall assessment is that the arguments presented within the manuscript are not really supported by the data and there are other explanations for results that are not equally explored. This is an observational study and there needs to be other supporting work to back up the idea that smaller colonies are being preferentially targeted. Moreover the authors fail to explore other reasons for the decline in overall feeding scars on porites. There are too many opinions inserted in the manuscript and overuse of emotive statements.

The argument in the manscript is not consistent as it oscillates between stating that massive porites are subject to less predation by parrotfish in a disturbed environment but that predation may be adversley affecting recruitment and rate of transitions of juvenile to adult porites.

line 72-73 What species are the main scrapers on porites corals - lumping all parrotfish together is not a powerful test

line 82-83 This statement has no supporting evidence and should be removed.

line 87 most corals have been lost is not a precise statement - do you mean they have gone extinct or severly reduced in abundance?

line92 dont need to add "stress- tolerant corals" your hypothesis is has reef degradation affected the intensity of parrotfish predation

line 94-95 "parrotfish are a growing threat to the persistence of corals on the GBR" is completely unsupported and this type of highly emotive statement has no place in the manuscript

line 101 - replace good with appropriate

line 124-125 this sentence is unclear - do you mean that photos were taken from a consistent distance above the substrate?

line 127 how did you decide on the number of points to analyse?

lines 140-146 this methodology needs to be more clearly explained or addressed as reviewer number 1 has pointed out.

lines 148 - 175 statistical tests need to be addressed as per reviewer 1's comments

line 234 You appear to be extrapolating way outside your study area. You have no evidence to support doing this. Given the degrees of patchiness that exist on reefs how do you know that the patterns observed in your study area are uniformly repeated across the entire reef

lines 293-299 You do not explore the equally plausible reason for decreased predation on porites is because parrotfish have found plenty of food around other than porites. The changes dont have to be manifest as increased numbers they can equally be a shift in feeding behaviour.

line 305 saying that predation moved in an unexpected direction is your opinion. As I have stated above there is another opinion that the change you saw was not unexpected. You need to stick to supported facts or else note that there are numerous possibilities none of which you have made a convincing case for.

lines 312-314 Why do you think predation was highest on the crest and reef flat. There is plenty of literature to support this pattern such as increased algae productivity?

line 318 "as one may expect" you must stop inserting your opinion as fact into the manuscript. just state the neutral case i.e. that you found scar numbers increased with colony size

lines 325-326 This would seem to be an avenue for further investigation. If there is targeted feeding of smaller colonies then all the characteristics of feeding on younger corals should be investigated. What about nutritional values?

lines 333-338 Juveniles of most species will experience higher mortality rates than adults this is not confined to porites corals. You would have to demonstrate that recruitment rates and juvenile abundance/densities declined below long-term rates to be able to support this statement.

lines 355-356 You have provided no evidence that there has been a decline in juvenile corals through predation by parrotfish.

lines 370-371 But you have not made the case that the level of predation on juvenile porites is any worse than previously. In fact it could be argued that access to increased algal/detrital resources as a result of disturbance has decreased the impacts of feeding scars as parrotfish have many other feeding opportunities

Reviewers' comments:

Reviewer's Responses to Questions

**Comments to the Author**

1. Is the manuscript technically sound, and do the data support the conclusions?

Reviewer #1: Partly

2. Has the statistical analysis been performed appropriately and rigorously? 

Reviewer #1: No

3. Have the authors made all data underlying the findings in their manuscript fully available?

Reviewer #1: Yes

4. Is the manuscript presented in an intelligible fashion and written in standard English?

Reviewer #1: Yes

5. Review Comments to the Author

Reviewer #1: This manuscript studied patterns of coral communities and feeding by coral-eating fishes (parrotfishes) in 2018 and compared these with the data in 2008. The comparison of these results between 2008 and 2018 was well described and the supporting data and analyses were fine. However, I am concern about the method and analysis used to determine Porites size and bites by parrotfishes and might include possible artefacts due to the survey methodology, as discussed below. Also, statistical analysis should be consistent, Information theory and Frequentist particularly should not be mixed in a single analysis for the same purpose.

In the photoquadrat survey, how the photos are taken and analysed should be more cleared. Looking at the large Porites colony in Fig. 4, it appears that the sides (which more likely to be bitten?) and the shadows of the large knots, were not being photographed. The lower density of bites on the larger Porites could simply be due to the increased blind spot of the camera as the size of the colony increases, so that many of the bites are not captured. In short, “surface area” of Porites in this manuscript seems to be more correctly “projected area”.

I wonder that the ratio of diameter to height changes with growth of the Porites colony. I think the small Porites colonies are often low in height relative to their diameter, whereas the large Porites colonies are high in height relative to their diameter. The height of the smaller colonies is only a few centimetres at most, so parrotfishes can easily access to the whole surface of small Porites colonies. However, a large Porites colony (Fig. 4 shows that the largest colony is about 70 cm high) might be several tens of centimetres high, could it be that parrotfishes, swimming near the seafloor, only bite the sides of the colony? If so, the density of bite marks will decrease. Additionally, I wonder that the relationship in Fig. 4A is ambiguous, whereas the relationships in Fig. 4B are all too apparent. This gap is particularly large for FLAT.

Is there no result about spatial distribution of Porites in each habitat? Also, the size density distribution of Porites colonies is also helpful. I think these information are important, because the higher the frequency of occurrence, the higher the encounter rate for parrotfishes and the more frequently being bitten. Usually, smaller coral colonies are more frequent than larger colonies. This can be analysed by adding the size frequency distribution of Porites colonies as an explanatory variable. cf. the frequency of each colony can be obtained using the 'density' function in R.

The illustration of the habitat types below in Figs. 2, 3, and 4 should be included in Fig.1 rather than showing in every figures. Also, the names of habitat type should be written rather than illustration in Figs. 3 and 4. The photos (Fig. 4c) also should be moved to Fig. 1.

The arrows in Fig. 3 should be avoided, since the arrows prevent readers from examining this figure without preconceptions.

In Table S1, weight of the AIC is usually called as “Akaike weights”, and “WAIC” usually represents Watanabe Akaike Information Criterion, which used in Bayesian model selections. To avoid misleading, different abbreviation should be used. I think “w” or “Wi” (“i” represents ID for each model) is more common for Akaike weights.

L171–173: Scar density should not be analysed by a GLM with a gamma distribution, since number of scars is a discrete countable variable. GLM with a negative binomial distribution should be used also in this case, but the surface area should be included as an “offset” term, i.e., adding “+ offset(log(SURFACE_AREA))” to the formula of GLM in R.

L174–175, Fig. 3 and Table S2: information theory (AIC model selection) and frequentist (Tukey post-hoc comparisons) should not be mixed in a single analysis. Information theory is also the best choice to perform multiple comparisons among groups, since its statistical power does not decrease with the number of comparisons unlike frequentist post-hoc tests. Just construct models of possible grouping from Slope to Back, and compare them with the AIC.

6. PLOS authors have the option to publish the peer review history of their article (what does this mean?). If published, this will include your full peer review and any attached files.

Reviewer #1: **Yes: **Naoki H. KUMAGAI

---

## [Author Response · Author response to Decision Letter 0]

19 Mar 2021

Thank you. We have revised the manuscript to incorporate the suggestions listed in the reviews and, where needed, we have clarified the text. Thank you for taking the time to review our manuscript. We appreciate the feedback provided to us.

---

## [Editor Report · Decision Letter 1]

13 Apr 2021

Parrotfish corallivory on stress-tolerant corals in the Anthropocene

PONE-D-20-34321R1

Dear Dr. Huertas,

We’re pleased to inform you that your manuscript has been judged scientifically suitable for publication and will be formally accepted for publication once it meets all outstanding technical requirements.

Kind regards,

Andrew Halford

Academic Editor

PLOS ONE

Additional Editor Comments (optional):

I thank the authors for their comprehensive response to the reviewers comments. The manuscript is much improved and now reads like a piece of considered science rather than an emotive statement.
---

## [Editor Report · Acceptance letter]

31 Aug 2021

PONE-D-20-34321R1 

Parrotfish corallivory on stress-tolerant corals in the Anthropocene 

Dear Dr. Huertas:

I'm pleased to inform you that your manuscript has been deemed suitable for publication in PLOS ONE. Congratulations! Your manuscript is now with our production department. 

Kind regards, 

on behalf of

Dr. Andrew Halford 

Academic Editor

PLOS ONE